# Nanoelectrode design from microminiaturized honeycomb monolith with ultrathin and stiff nanoscaffold for high-energy micro-supercapacitors

Zhendong Lei[1,2,5], Long Liu[1,5], Huaping Zhao[1]*, Feng Liang [3]*, Shilei Chang[3], Lei Li[4], Yong Zhang [2], Zhan Lin[4]*, Jörg Kröger [1] & Yong Lei [1]*

Downsizing the cell size of honeycomb monoliths to nanoscale would offer high freedom of nanostructure design beyond their capability for broad applications in different fields. However, the microminiaturization of honeycomb monoliths remains a challenge. Here, we report the fabrication of microminiaturized honeycomb monoliths—honeycomb alumina nanoscaffold—and thus as a robust nanostructuring platform to assemble active materials for micro-supercapacitors. The representative honeycomb alumina nanoscaffold with hexagonal cell arrangement and 400 nm inter-cell spacing has an ultrathin but stiff nanoscaffold with only $16 \pm 2$ nm cell-wall-thickness, resulting in a cell density of $4.65 \times 10^9$ cells per square inch, a surface area enhancement factor of 240, and a relative density of 0.0784. These features allow nanoelectrodes based on honeycomb alumina nanoscaffold synergizing both effective ion migration and ample electroactive surface area within limited footprint. A micro-supercapacitor is finally constructed and exhibits record high performance, suggesting the feasibility of the current design for energy storage devices.

---

[1] Institute of Physics and IMN MacroNano®, Ilmenau University of Technology, Ilmenau, Germany. [2] NUS Graduate School for Integrative Sciences and Engineering, National University of Singapore, Singapore, Singapore. [3] Faculty of Metallurgical and Energy Engineering, Kunming University of Science and Technology, Kunming, China. [4] School of Chemical Engineering and Light Industry, Guangdong University of Technology, Guangzhou, China. [5] These authors contributed equally: Zhendong Lei, Long Liu. *email: huaping.zhao@tu-ilmenau.de; liangfeng@kust.edu.cn; zhanlin@gdut.edu.cn; yong.lei@tu-ilmenau.de

Honeycomb monoliths (HMs) are a kind of cellular materials with two-dimensional cell configuration consisting of parallel and straight channels extended throughout the body[1]. HMs have been widely exploited as catalyst supporters in different gaseous reactor applications such as chemical and refining processes, catalytic combustion, ozone abatement, and photocatalytic air purification[2–7]. As catalyst supporters, HMs have higher surface-to-volume ratio comparing to that of planar substrates for providing high contact efficiencies between the supported catalysts and the gaseous reactants, meanwhile their parallel and straight channels facilitate mass transport of gas with low diffusion resistance. Besides these gas-phase catalysis applications, the cellular and homogeneous structure of the HM presents potential opportunities for other functional applications. For example, it could be a desirable platform to assemble an electroactive electrode for electrochemical devices (e.g., batteries, supercapacitors, fuel cells, water electrolyzer, and electrochemical sensors), in which an efficient electrochemical reaction requires both high specific surface area of the electrode and favorable ionic transport within the electrode[8–11]. Despite these merits, so far HMs have not been widely utilized for electrochemical applications mainly due to the bulky structure of the existing HMs. The present electrode design based on HMs makes electrochemical devices cumbersome (especially the device volume), and meanwhile the macrostructural channels of HMs require an impractically large amount of electrolytes to fill up all the channels in order to ensure a sufficient contact between the supported electroactive materials and the electrolyte, which is different from gas-phase catalysis applications. Moreover, the specific surface area of the conventional HMs, although it is higher than that of planar substrate counterparts, is far from sufficient to satisfy the requirements of electrochemical applications. Currently, the best reported HM can only attain $1.6 \times 10^4$ cells per square inch (cpsi) and about 3 μm in cell wall thickness[12], with a specific surface area not being comparable with that of widely used nanoelectrodes (i.e., mainly arrays of nanowires or nanotubes). However, most of those nanoelectrodes suffer from the limit of aspect ratio to maintain their highly oriented nature by avoiding the agglomeration since nanowires and nanotubes with high aspect ratio would prefer to form into many dense clusters, and consequently are difficult to simultaneously satisfy both high specific surface area and low ion transport resistance. To this end, downsizing the cell size (e.g., channel diameter and cell wall thickness) of the conventional HMs to nanoscale shall be a solution to efficiently address the challenges for extending the application potentials of HMs. Nevertheless, the microminiaturization of the HMs encounters technological difficulties in creating exactly parallel and straight nanoscale channels over a large area by all reported HM fabrication approaches whatever in industry or in laboratory, and meanwhile faces the challenge to guarantee that the microminiaturized HM would hold the similar excellent mechanical stabilities as the conventional HM.

In this article, microminiaturized HMs with ultrathin and stiff nanoscaffold—honeycomb alumina nanoscaffold (HAN)—is realized for the first time. Specifically, a large-scale HAN is fabricated by using a nanoindentation-anodization-etching process with a high-purity aluminum foil. A representative HAN is actually a microminiaturized HM with hexagonal cell arrangement, 400 nm inter-cell spacing and only $16 \pm 2$ nm cell wall thickness. Impressively, the cell density of the HAN reaches $4.65 \times 10^9$ cpsi that is five orders of magnitude higher than that of the reported highest value. Meanwhile, the HAN with such ultrathin cell wall is stiff and can sustain surprisingly high mechanical stability identified by an experimental nanoindentation. Such robust HAN offers a stable nanostructuring platform to assemble electroactive materials for micro-supercapacitors

(MSCs). The insulating HAN is retained in the nanoelectrodes and thus endows the nanoelectrodes with vertically aligned and robustly stable nanoporous structure to simultaneously achieve both effective ion migration and ample electroactive surface area within limited footprint. As a result, a MSC constructed with the HAN-based nanoelectrodes exhibits record high-energy performance among the reported micro-supercapacitors. The maximum capacitance of the MSC reaches 128 mF cm$^{-2}$ at a current density of 0.5 mA cm$^{-2}$, and the peak energy and power densities are 160 μWh cm$^{-2}$ and 40 mW cm$^{-2}$, respectively.

## Results

**Formation and structure of HAN**. A schematic of the farbication process of HAN, and the corresponding scanning electron microscope (SEM) images, are shown in Fig. 1a–h. Briefly, the fabrication of HAN includes nanoindentation, anodization, and a two-step etching process (Fig. 1a). The fabrication process begins with the anodization of a surface-nanopatterned aluminum substrate to obtain nanoporous alumina ($Al_2O_3$) with a hexagonal cell arrangement and a cell periodicity of 400 nm. Notably, the as-prepared nanoporous alumina has a double-layer cell structure (Fig. 1b), with an acid acid anion-contaminated $Al_2O_3$ thick layer (in dark-gray color with the thickness of $90 \pm 2$ nm) adjacent to the cell center and a relatively pure $Al_2O_3$ thin layer (in light-gray color with the thickness of $21 \pm 2$ nm) remote from the cell center[12]. By applying a precisely controlled two-step etching in aqueous $H_3PO_4$ solutions, the acid anion-contaminated $Al_2O_3$ thick layer are easy to be totally dissolved and meanwhile the pure $Al_2O_3$ thin layer is partially etched. Finally, a honeycomb nanoscaffold consisting of only an ultrathin layer of pure $Al_2O_3$ is left with the cell wall thickness of $16 \pm 2$ nm (Fig. 1c), which is a HAN (i.e., a microminiaturized HM). Figure 1d schematically shows the evolution of the HAN through the second etching process. In the case of the HAN with hexagonal cell arrangement and 400 nm inter-cell spacing, the cell density is calculated to be $4.65 \times 10^9$ cpsi that is five orders of magnitude higher than that of the reported highest value[13]. The cross-sectional SEM image of a HAN with cell depth of about 25 μm confirms the straight and stable nanoporous structure of the HAN (Fig. 1e). Noting that the cell depth of the HAN depends on the anodization time, by tuning the anodization time, the cell depth of the HAN can be adjusted from hundreds of nanometers to hundreds of micrometers. The surface area enhancement factor for an above-mentioned HAN with the 25-μm-deep cell is calculated to be 240. And, importantly, the relative density of HAN is calculated to be only 0.0784, corresponding to its porosity of 0.9216. When applying such HAN as a scaffold for designing electrodes and/or devices, the high specific surface area and the lower "dead volume" (i.e., the proportion of the insulating $Al_2O_3$ to the whole electrode) in electrodes/devices will be simultaneously satisfied, benefiting to accomplishing high device performance. Not limited to hexagonal cell arrangement with 400 nm inter-cell spacing, two other HANs also have been produced through tuning the nano-patterns on the surface of aluminum foils generated by nanoindentation, one with hexagonal cell arrangement but 800 nm inter-cell spacing (Fig. 1f) and the other with 400 nm inter-cell spacing but square cell arrangement (Fig. 1g). Note that the HAN can be fabricated over large area with almost no structural defects (Fig. 1h and Supplementary Fig. 1a–b), which is also an important advantage of the HAN for electrode designing.

Generally, a compressive stack pressure typically in the range of 0.1–10 MPa is applied during the assembly of batteries (coin- and punch-cell) and supercapacitors for the purpose of maintaining intimate contact between the electrodes and the current collectors as well as preventing active materials delamination and deformation in

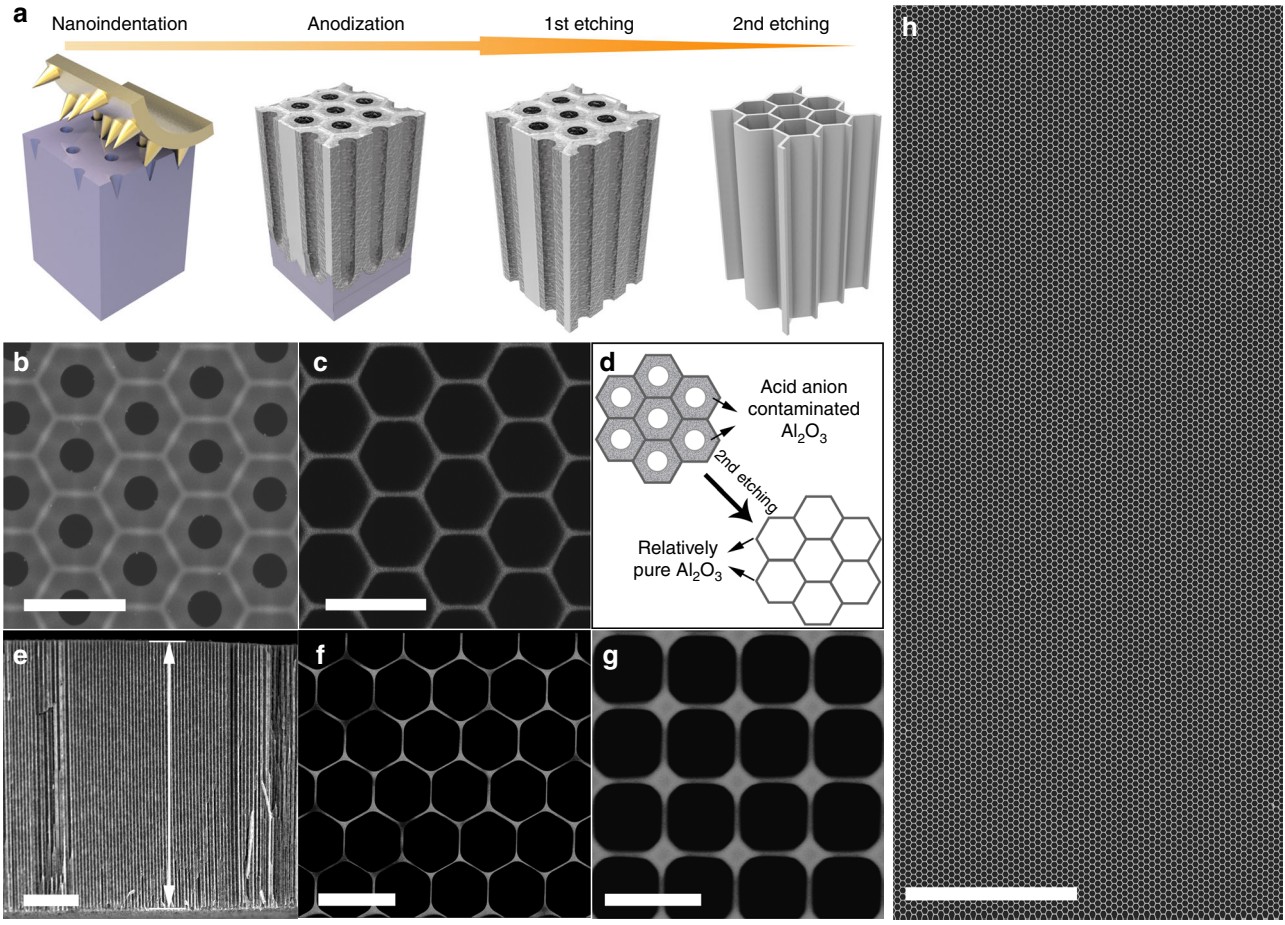

**Fig. 1 Fabrication and structure of HAN. a** Illustration of the HAN fabrication process. **b** SEM image of nanoporous alumina after anodization process revealing the double-layer cell structure. **c** SEM image of HAN with hexagonal cell arrangement and 400 nm inter-cell spacing showing the structure of HAN with ultrathin wall of $16 \pm 2$ nm. **d** Schematic of the evolution of the HAN by the second etching process. **e** Cross-sectional view SEM image of an HAN indicating the cell depth to be about 25 μm. **f** SEM image of HAN with hexagonal cell arrangement and 800 nm inter-cell spacing. The wall thickness of the cell is $18 \pm 2$ nm. **g** SEM image of HAN with square cell arrangement and 400 nm inter-cell spacing. The wall thickness of the cell is $12 \pm 2$ nm. **h** Large-scale SEM image showing the uniform, stable and defect-free structure of the HAN with hexagonal cell arrangement and 400 nm inter-cell spacing. Scale bar: 500 nm (**b**); 500 nm (**c**); 5 μm (**e**); 1 μm (**f**); 500 nm (**g**); 10 μm (**h**).

the electrodes during operation[14]. Obviously, such high extrusion pressure would bring forward the critical challenges and requirements to the mechanical stabilities of the nanoelectrodes for supercapacitors and batteries. As known, the cellular solids with honeycombs are much stiffer and stronger to manifest high compressive strength when loaded at the out-of-plane direction (i.e., along the cell axis) than at the in-plane direction[15,16], thus nanoindentation experiments have been further conducted for characterizing the nanomechanical properties of the HAN. The Young's modulus of the HAN with hexagonal cell arrangement, 400 nm inter-cell spacing and only 16 nm cell wall thickness are obtained in the range of 1.41–3.22 GPa measured at five different positions (Supplementary Fig. 1c). The excellent mechanical performance of the HAN should be attributed to not only the high hardness of the pure $Al_2O_3$ and also the anisotropic honeycomb microarchitectures of the HAN[17–20], and it would be a significant advantage when applying as nanostructuring platform to assemble electroactive materials especially for supercapacitors and batteries, because the HAN can play the part of a rigid keel in the nanoelectrodes to sustain the mechanical extrusion during the device assembly but maintain the structural features at the nanoscale of the electrodes, which are inherited from the HAN, for finally accomplishing nanoelectrodes with both high specific surface area and low ion transport resistance for efficient electrochemical energy storage.

**Fabrication and characterizations of HAN@SnO₂ as nanostructured current collectors.** As aforementioned, the HAN is a kind of microminiaturized HMs with much higher cell density than the conventional HMs, and the cellular and homogeneous structure of the HAN presents opportunities as nanostructuring platform to assemble electroactive materials for electrochemical devices. Given the insulating nature of the HAN, the first step to design nanoelectrodes with the HAN for electrochemical applications is to convert the insulating HAN to be a nanostructured current collector. As shown in Fig. 2a, a 12-nm-thick tin oxide ($SnO_2$) layer was conformally coated onto the HAN with hexagonal cell arrangement and 400 nm inter-cell spacing (Fig. 1c) through atomic layer deposition (ALD). Note that a thick layer of nickel (~10 μm) as the supporting substrate was electrochemically deposited onto the top of the HAN before the conduction of the ALD process. The $SnO_2$-coated HAN (denoted as HAN@$SnO_2$) preserves the original honeycomb features with the cell wall thickness increasing from $16 \pm 2$ to $40 \pm 2$ nm. To verify the possibility of HAN@$SnO_2$ as nanostructured current collectors, a

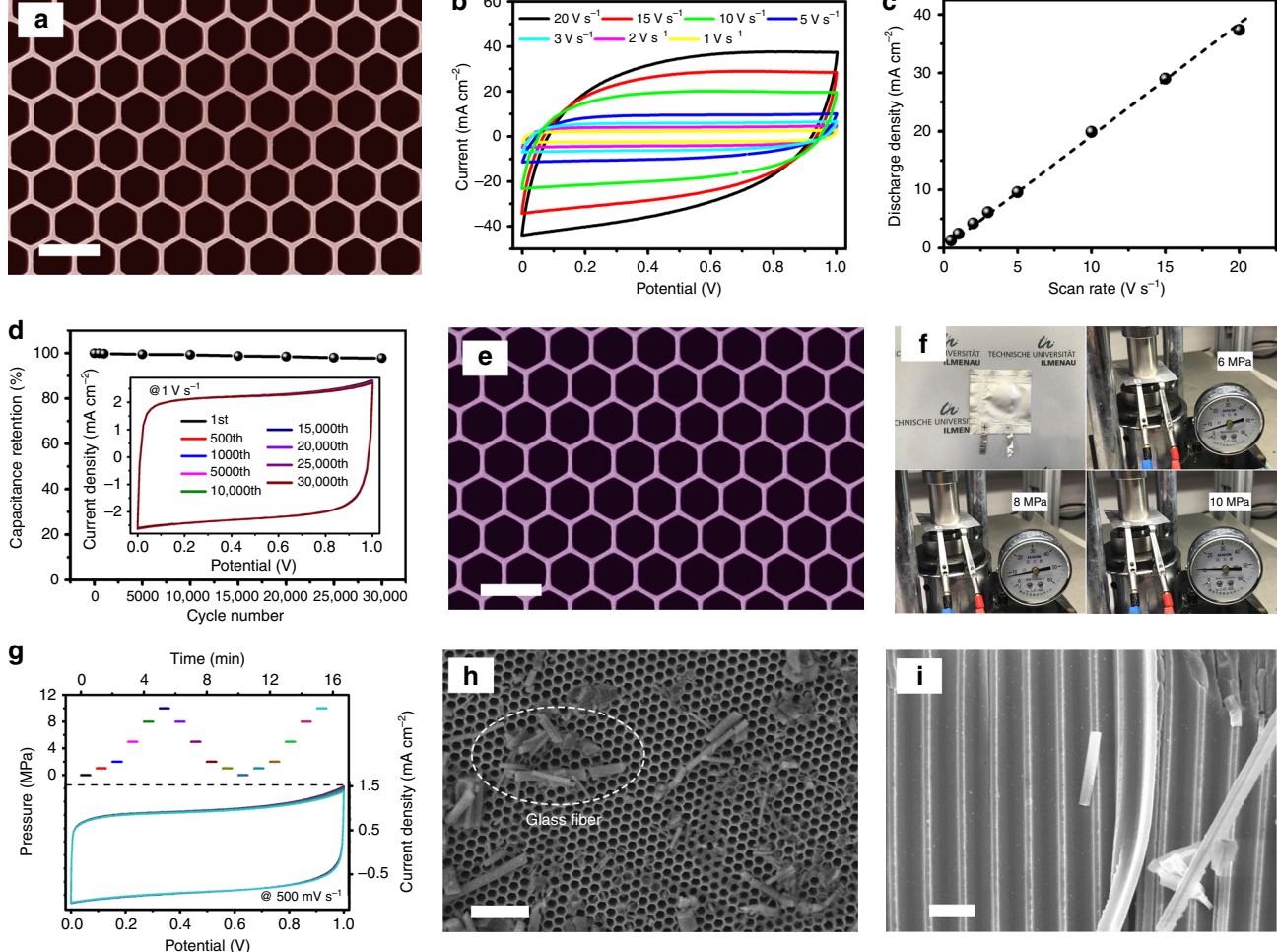

**Fig. 2 Characterizations of HAN@SnO₂ as nanostructured current collectors. a** Top-view SEM image of HAN@SnO₂. **b** CV curves of HAN@SnO₂//HAN@SnO₂ device at different scan rates. **c** The discharge current as function of the scan rates based on the CV curves of HAN@SnO₂//HAN@SnO₂ device at different scan rates. **d** Cycling stability of HAN@SnO₂//HAN@SnO₂ device tested at a scan rate of 1 V s⁻¹. **e** Top-view SEM image of HAN@SnO₂ electrode after 30,000 CV cycles. **f** Photographs of HAN@SnO₂//HAN@SnO₂ device tested under different mechanical extrusion pressure. **g** CV profiles of HAN@SnO₂//HAN@SnO₂ device measured at a scan rate of 500 mV s⁻¹ under different mechanical extrusion pressure. **h** Top-view, and **i** cross-sectional SEM image of HAN@SnO₂ electrode after CV test under a mechanical extrusion pressure of 10 MPa, respectively. Scale bar: 500 nm (**a**, **e**); 2 μm (**h**); 500 nm (**i**).

symmetric supercapacitor based on two HAN@SnO₂ electrodes, HAN@SnO₂//HAN@SnO₂, were assembled and characterized since SnO₂ is also one kind of electroactive materials for supercapacitors. Rate performance of supercapacitors, especially under high scan rate, could indirectly reflect the ionic and electronic transport behavior of electrodes. The cyclic voltammetry (CV) curves of the device at different scan rates are shown in Fig. 2b, and all have a quasi-rectangular shape even at a high scan rate of 20 V s⁻¹, which indicates the typical double-layer capacitive behavior of the device. Figure 2c reveals the corresponding variation of the discharge current density at different scan rates. Clearly, the discharge currents keep a linear relationship upon scan rates nearly until 20 V s⁻¹, suggesting the high instantaneous power characteristics of the device. The representative Nyquist plot of the device, and the corresponding expanded view at the high-frequency region, again reveals the capacitive behavior even at high frequencies that are attributed to the fully accessible surface area of the electrodes for electrolyte ions adsorption/desorption (Supplementary Fig. 2). Moreover, the high-frequency semicircle reflects the small charge transfer resistance ($R_{ct}$) of only 1.8 Ω in HAN@SnO₂//HAN@SnO₂ device, also contributing to realize high-rate performance. Therefore, high-rate performance

should be attributed not only to the rapid mass transport of ions but also to the efficient charge transport and collection in HAN@SnO₂ nanoelectrodes during the ultrafast charge-discharge process[21].

In addition, a nearly 98% of the initial performance was retained after continued 30,000 charge/discharge cycles at a scan rate of 1 V s⁻¹ (Fig. 2d) and no obvious structural changes and/or damages can be found for the electrodes when comparing the SEM images of the HAN@SnO₂ electrodes before and after long-termed cycling (Fig. 2a, e, respectively). The high stability in both electrochemical performance and electrode structure should be attributed to the existence of the HAN in the nanoelectrodes to functionalize as mechanically robust keel. On the other hand, it is well-known that the mechanical extrusion is an essential step during the device assembly of supercapacitors and batteries[14], however, at the same time it would generate a destructive impact on nanoelectrodes. For example, the collapse of nanostructures and then the block of ionic transport pathway will subsequently result in unexpectedly device performance degradation compared to the result of three-electrode configuration. Attributing to the high stiffness of Al₂O₃, the monolithic structure of HAN without structural defects over a large area and the high Young's modulus

of the HAN, HAN@SnO$_2$ electrodes are endowed with high compression strength to sufficiently withstand the mechanical extrusion during the device assembly without collapse or deformation of the nanoporous structure. As a result, HAN@SnO$_2$ electrodes could remain their initial integrated structure with vertically aligned nanoporous structure and finally, the HAN@SnO$_2$//HAN@SnO$_2$ device still works very well without deteriorating the capacitive behavior even under an applied mechanical extrusion pressure up to 10 MPa (Fig. 2f, g). The constant CV profiles indicate that there is nearly no change in the ion-accessible surface area and the ion transport resistance under the continuous mechanical extrusion. Furthermore, the SEM images (Fig. 2h, i) reveal that the HAN@SnO$_2$ electrodes still remain the original nanoporous features after the electrochemical test under the continuous mechanical extrusion. Note that the residues in Fig. 2h, i is the residual glass fibers from a glass microfiber filter that was used as separator, and partials of the glass microfiber filter were damaged and fallen off during the disassembly of the HAN@SnO$_2$//HAN@SnO$_2$ device. This result further reveals the excellent mechanical stability of the HAN that can efficiently sustain the mechanical extrusion pressure during the device assembly process to maintain the integrality of the nanoelectrodes. Overall, the excellent electrochemical performance and the superior structural stability verify the great capability of HAN@SnO$_2$ as desirable nanostructured current collectors to design nanoelectrodes for batteries and supercapacitors.

**Assembly and electrochemical performance of symmetric MSCs.** Following the electrochemical characterization of HAN@SnO$_2$ as nanostructured current collectors, a layer of pseudocapacitive materials was further electrochemically deposited with HAN@SnO$_2$ as nanostructured current collectors to produce nanoelectrodes for MSCs, which are magnesium oxide (MnO$_2$) coated HAN@SnO$_2$ (HAN@SnO$_2$@MnO$_2$) and polypyrrole (PPy) coated HAN@SnO$_2$ (HAN@SnO$_2$@PPy) nanoelectrodes, respectively. The pore depth of HAN in both two nanoelectrodes was 25 μm, and the mass loading of MnO$_2$ and PPy was 0.65 and 1.32 mg cm$^{-2}$, respectively. The detailed cross-sectional SEM images of different sections (Supplementary Fig. 3) and the corresponding energy dispersive X-ray spectroscopy (EDX) elemental mapping results (Supplementary Figs. 4–5) clarify that the cell wall of HAN@SnO$_2$ have been coated with PPy and MnO$_2$, respectively. Thereafter, MSCs with symmetric device configuration were assembled through stacking two same HAN-based nanoelectrodes separated by a glass microfiber filter and with 1.0 M Na$_2$SO$_4$ aqueous solution as electrolyte. Note that the footprint of all stacked MSCs is 0.5 cm$^2$, which is equal to that of single electrode. Figure 3a is the CV curves of the assembled HAN@SnO$_2$@MnO$_2$//HAN@SnO$_2$@MnO$_2$ MSCs over a wide range of scan rates from 2 to 500 mV s$^{-1}$ in a potential range of 0–0.8 V. Obviously, the CV curves at different scan rates exhibit a symmetrically rectangular shape, indicating an ideally capacitive behavior and fast charge–discharge characteristics of MSCs. The galvanostatic charge–discharge (GCD) curves of HAN@SnO$_2$@MnO$_2$//HAN@SnO$_2$@MnO$_2$ MSCs at different current densities are shown in Fig. 3b. These charging curves are very symmetric to the discharge counterparts, again indicating the excellent capacitive behaviors in MSCs. When depositing MnO$_2$, the layer thickness of MnO$_2$ was kept the same in all HAN@SnO$_2$@MnO$_2$ electrodes, thus electrodes with deeper HAN have higher MnO$_2$ mass loading. Figure 3c illustrates the device areal capacitance of MSCs with different pore depth of original HAN and MnO$_2$ mass loading as a function of scan rates. The device capacitance depends strongly on the pore depth of original HAN

and the subsequent changes of MnO$_2$ mass loading of HAN@SnO$_2$@MnO$_2$ electrodes. For HAN@SnO$_2$@MnO$_2$ electrodes consisting of 25-μm-pore-deep HAN, the MnO$_2$ mass loading was 0.65 mg cm$^{-2}$, consequently resulting in highest device capacitance of 137 mF cm$^{-2}$ at a scan rate of 2 mV s$^{-1}$ (121 mF cm$^{-2}$ at a current density of 0.2 mA cm$^{-2}$, Supplementary Fig. 6). Benefiting from the vertically aligned nanoporous structure features, these symmetric MSCs have good rate performance, i.e., the capacitance of MSCs based on HAN@SnO$_2$@MnO$_2$ electrodes with 25-μm-deep HAN remains to be 47 mF cm$^{-2}$ when the scan rate increasing from 10 to 2,000 mV s$^{-1}$. Additionally, the HAN@SnO$_2$@PPy//HAN@SnO$_2$@PPy MSCs have the similar electrochemical performance to that of HAN@SnO$_2$@MnO$_2$//HAN@SnO$_2$@MnO$_2$ MSCs except the different operating potential window that is from −0.8 to 0 V. Figure 3d, e is the CV and GCD curves of HAN@SnO$_2$@PPy//HAN@SnO$_2$@PPy MSCs, respectively. The pore depth of original HAN in electrodes was 25 μm and the corresponding PPy mass loading was 1.32 mg cm$^{-2}$, and in this case, the maximum device capacitance of HAN@SnO$_2$@PPy//HAN@SnO$_2$@PPy MSCs reaches 124 mF cm$^{-2}$ at a scan rate of 10 mV s$^{-1}$ (158 mF cm$^{-2}$ at a current density of 0.2 mA cm$^{-2}$, Supplementary Fig. 6).

By contrast, HAN@SnO$_2$//HAN@SnO$_2$ MSCs with the same footprint (0.5 cm$^2$) were also assembled and characterized to further understand the role of SnO$_2$ layer in HAN@SnO$_2$ based pseudocapacitive electrodes. It can be concluded from Fig. 3c, f that the negligible capacitance (<4 mF cm$^{-2}$) of HAN@SnO$_2$//HAN@SnO$_2$ MSCs, in which the pore depth of HAN in all electrodes was 25 μm, suggests the function of SnO$_2$ layer in HAN@SnO$_2$ based pseudocapacitive electrodes mostly for charge transport rather than charge storage. Furthermore, electrochemical impendence spectroscopy (EIS) was performed in order to clearly understand the role of HAN@SnO$_2$ as nanostructured current collectors. Figure 4a is the Nyquist plots of HAN@SnO$_2$//HAN@SnO$_2$ MSCs with different pore depth of original HAN (i.e., 5, 16 and 25 μm, respectively). All MSCs have very similar equivalent series resistances (ESR), however, there is a significant raise in $R_{ct}$. The gradually increased $R_{ct}$ should be the inevitable result arising from the limited electrical conductivity of SnO$_2$, which is much lower than that of metals. The intrinsic electrical resistance of SnO$_2$ leads the total resistance of HAN@SnO$_2$//HAN@SnO$_2$ MSCs to be progressively increased accompanying with HAN pore depth increase, and then consequently results in an increase in $R_{ct}$. With respect to HAN@SnO$_2$@MnO$_2$//HAN@SnO$_2$@MnO$_2$ and HAN@SnO$_2$@PPy//HAN@SnO$_2$@PPy MSCs (Fig. 4b, c, respectively), all exhibit similar Nyquist curves, including a semicircle in the high-frequency region and a nearly vertical line in the low-frequency region. In the high-frequency region, the diameters of the semicircle for HAN@SnO$_2$@MnO$_2$//HAN@SnO$_2$@MnO$_2$ MSCs are bigger than those of HAN@SnO$_2$@PPy//HAN@SnO$_2$@PPy MSCs, indicating a higher $R_{ct}$ of HAN@SnO$_2$@MnO$_2$//HAN@SnO$_2$@MnO$_2$ MSCs. Additionally, the higher $R_{ct}$ of HAN@SnO$_2$@MnO$_2$//HAN@SnO$_2$@MnO$_2$ MSCs are also evidenced by the more obvious potential drops of GCD curves than those of HAN@SnO$_2$@PPy//HAN@SnO$_2$@PPy MSCs (Fig. 3b, e). In addition, both HAN@SnO$_2$@MnO$_2$//HAN@SnO$_2$@MnO$_2$ and HAN@SnO$_2$@PPy//HAN@SnO$_2$@PPy MSCs have the similar ESR as those of HAN@SnO$_2$//HAN@SnO$_2$ MSCs. The similar ESR suggests the negligible influence on the series bulk resistance with the use of HAN@SnO$_2$ as nanostructured current collectors. In addition, the smaller ESR in HAN@SnO$_2$@PPy//HAN@SnO$_2$@PPy MSCs (Fig. 4c) could be attributed to the higher electrical conductivity of PPy than that of MnO$_2$. Figure 4d summarizes the $R_{ct}$ of all HAN-based MSCs. For HAN@SnO$_2$@MnO$_2$//HAN@SnO$_2$@MnO$_2$ MSCs, the $R_{ct}$ are all higher than those of

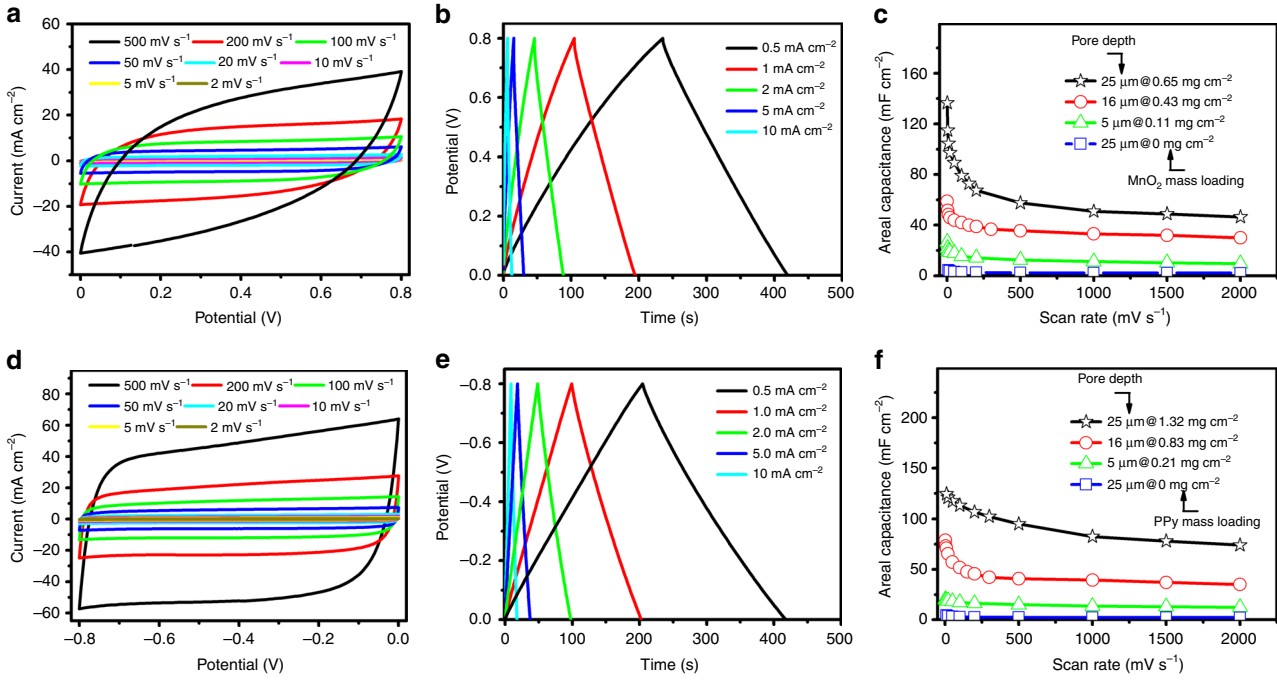

**Fig. 3 Electrochemical performance of symmetric MSCs based on HAN-based nanoelectrodes. a** CV curves at different scan rates, **b** GCD profiles at different current densities, and **c** device areal capacitance as a function of scan rates of HAN@SnO$_2$@MnO$_2$//HAN@SnO$_2$@MnO$_2$ MSCs, respectively. **d** CV curves at different scan rates, **e** GCD profiles at different current densities, and **f** device areal capacitance as a function of scan rates of HAN@SnO$_2$@PPy//HAN@SnO$_2$@PPy MSCs, respectively.

HAN@SnO$_2$//HAN@SnO$_2$ MSCs. Besides the increased electrical resistance in HAN@SnO$_2$ current collectors, the gradual raise in $R_{ct}$ of HAN@SnO$_2$@MnO$_2$//HAN@SnO$_2$@MnO$_2$ MSCs should be due to the low intrinsic conductivity of MnO$_2$. The low conductive MnO$_2$ layer makes the charge transport in HAN@SnO$_2$@MnO$_2$ electrodes mainly rely on the conductive SnO$_2$ layer, which results in the exclusive pathway for charge transportation. On the contrary, the highly conductive PPy layer could not only store charges but also partially contribute to the charge transport for improving the charge transport efficiency in HAN@SnO$_2$@PPy electrodes, resulting in lower $R_{ct}$ of HAN@SnO$_2$@PPy//HAN@SnO$_2$@PPy MSCs than those in both HAN@SnO$_2$//HAN@SnO$_2$ and HAN@SnO$_2$@MnO$_2$//HAN@SnO$_2$@MnO$_2$ MSCs. Furthermore, fast ion transport at the interface between electrode and electrolyte could be evident by the inconspicuous Warburg region. In the low-frequency region (Supplementary Fig. 7), all HAN-based MSCs exhibit an almost vertical line, representing the promising permeability for electrolyte infiltration and ion diffusion to access the surface of pseudocapacitive materials, and to create more active sites for electrochemical reactions to store more charges.

**Construction and performance evaluation of asymmetric MSCs.** The areal energy density is directly proportional to the areal capacitance value and the square of the cell voltage. Nevertheless, the energy density of supercapacitors is restricted by the use of pseudocapacitive materials with a narrow potential window. An asymmetric device configuration affords the opportunity for the expansion of the operating potential window of pseudocapacitors, and subsequently accomplishes the enhanced energy and power densities. Asymmetric MSCs were therefore assembled by using HAN@SnO$_2$@MnO$_2$ as positive electrode and HAN@SnO$_2$@PPy as negative electrode. The two electrodes were stacked together with a glass microfiber filter as separator and 1.0 M Na$_2$SO$_4$ aqueous solution as electrolyte. As

the aforementioned symmetric MSCs, the footprint of HAN@SnO$_2$@MnO$_2$//HAN@SnO$_2$@PPy MSCs was also 0.5 cm$^2$. Note that a charge balance between the two electrodes was accomplished by controlling the deposition time of MnO$_2$ at the positive electrode and the thickness of the PPy film at the negative electrode. Figure 5a outlines the working potential windows of each symmetric MSCs individually, −0.8–0 V for HAN@SnO$_2$@PPy//HAN@SnO$_2$@PPy MSCs and 0–0.8 V for HAN@SnO$_2$@MnO$_2$//HAN@SnO$_2$@MnO$_2$ MSCs, so that the asymmetric MSCs by integrating HAN@SnO$_2$@PPy and HAN@SnO$_2$@MnO$_2$ into single MSCs will result in an increased operating cell voltage up to 1.6 V. As shown in Fig. 5b, the asymmetric MSCs with 16-μm-pore-deep HAN in all electrodes work efficiently in the range from 0.8 to 1.6 V. Figure 5c, d is the corresponding CV and GCD curves of asymmetric MSCs in a potential range of 0–1.6 V, respectively. The nearly rectangular CV curves and the highly triangular charge–discharge profiles demonstrate an ideally capacitive behavior and the fast charge–discharge characteristics of the asymmetric MSCs. Particularly, the CV curves retain their rectangular shape without apparent distortions when increasing scan rates up to a high rate of 1000 mV s$^{-1}$, indicating the extraordinary high-rate performance of asymmetric MSCs. And a highest device capacitance of 147 mF cm$^{-2}$ has been achieved at a scan rate of 10 mV s$^{-1}$ and still remain 80 mF cm$^{-2}$ at the scan rate of 1000 mV s$^{-1}$ (Supplementary Fig. 6). The maximum capacity of this device could reach 186 mF cm$^{-2}$ at a current density of 0.2 mA cm$^{-2}$ (Supplementary Fig. 8), and is one of the highest values among the reported MSCs (Supplementary Table 1). Moreover, the asymmetric MSCs have an excellent cyclic capability of 87% of its original capacity after continued 30,000 charge/discharge cycles tested at a current density of 20 mA cm$^{-2}$ and with nearly 100% Coulombic efficiency (Fig. 5f). The slight performance degradation should be mainly due to MnO$_2$, as evidenced by the obvious morphological changes after cycling (Supplementary Fig. 9).

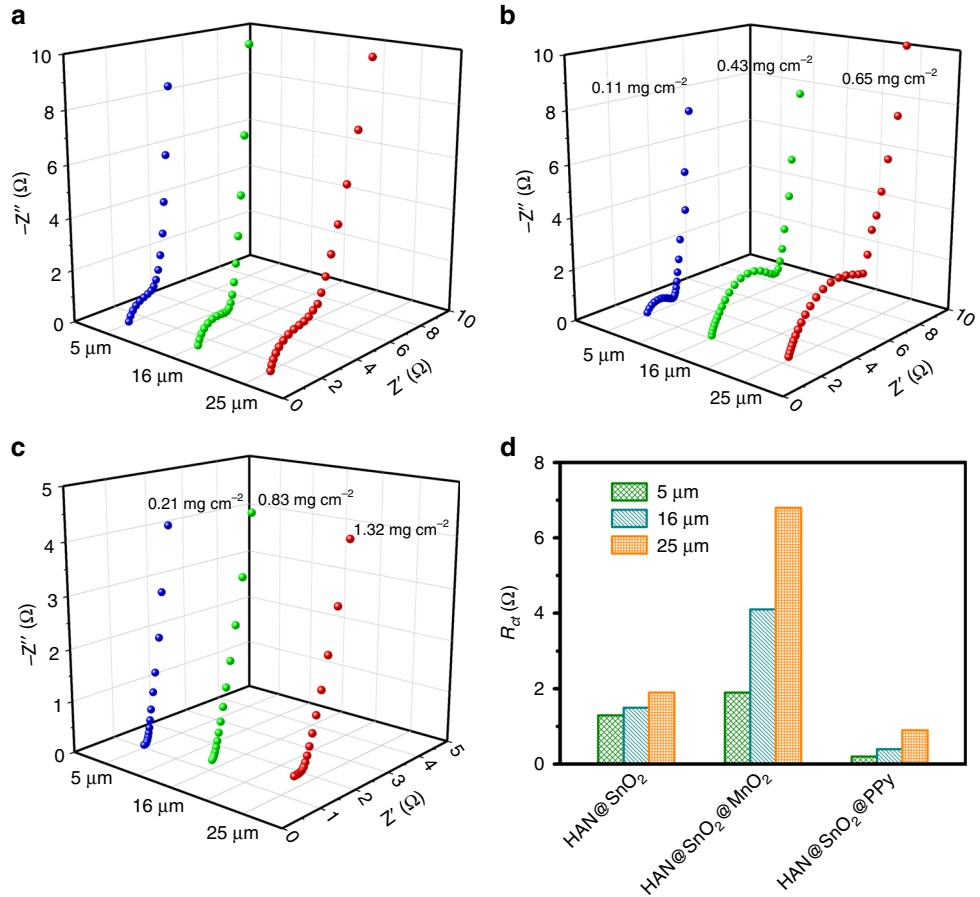

**Fig. 4 Electrochemical impedance properties comparison.** Nyquist plots of **a** HAN@SnO$_2$//HAN@SnO$_2$, **b** HAN@SnO$_2$@MnO$_2$//HAN@SnO$_2$@MnO$_2$, and **c** HAN@SnO$_2$@PPy//HAN@SnO$_2$@PPy MSCs with different pore depths of HAN. **d** Comparison of the charge transport resistance ($R_{ct}$) of different HAN-based MSCs.

When further replacing aqueous electrolyte with the ionic liquid electrolyte (1-ethyl-3-methylimidazolium bis(trifluoro-methylsulfonyl)imide, EMIM-TFSI), the asymmetric MSCs can work efficiently in an enlarged potential range of 0–3.0 V (Fig. 5g, h). The distorted CV and GCD curves arise from the high viscosity and large charge transfer resistance of the electrolytes because of the large ion size of the ionic liquids[22,23]. The device capacitance reaches 128 mF cm$^{-2}$ at a current density of 0.5 mA cm$^{-2}$ (or 162 mF cm$^{-2}$ at a scan rate of 10 mV s$^{-1}$). When the current density is increased from 0.5 to 5 mA cm$^{-2}$, the capacitance drops to 93 mF cm$^{-2}$ (72% retention) and it recovers to 123 mF cm$^{-2}$ (96% retention) with the current density again being decreased to 0.5 mA cm$^{-2}$ (Fig. 5i), revealing the excellent rate performance of the asymmetric MSCs. In addition, the asymmetric MSCs with ionic liquid electrolyte still exhibit remarkable cycling stability with 82.5% of initial capacitance at 20 mA cm$^{-2}$ over a potential window of 0–3.0 V withstanding continued 10,000 cycles with high Coulombic efficiency of nearly 100% and slight decay in CV curves at 100 mV s$^{-1}$ before and after cycling (Supplementary Fig. 10). Figure 6a presents the areal energy performance metrics of one representative HAN@S-nO$_2$@MnO$_2$//HAN@SnO$_2$@PPy asymmetric MSCs with EMIM-TFSI ionic liquid electrolyte. The maximum capacitance of the MSCs reaches 128 mF cm$^{-2}$ at a current density of 0.5 mA cm$^{-2}$, and the peak energy and power densities are 160 µWh cm$^{-2}$ and 40 mW cm$^{-2}$, respectively. Remarkably, the energy storage performance of the asymmetric MSCs with HAN-based nanoelectrodes is among the best comprehensive performance of the reported stacked MSCs (Fig. 6b)[21,24–34]. In particular, the peak energy density of the stacked asymmetric MSCs in this work is roughly fourfold that of the carbide-derived-carbons (CDC) based stacked MSCs (~40 µWh cm$^{-2}$) but with a similar peak power density[24]. Additionally, the areal energy density of the asymmetric MSCs with EMIM-TFSI electrolyte is even comparable with that of some state-of-the-art 3D micro-batteries but with much higher areal power density[35,36].

## Discussion

The insufficient energy of MSCs is their crucial flaw compared with micro-batteries and remains a major challenge to overcome. This work signifies the capability of the HAN as a promising nanostructuring platform to assemble pseudocapacitive materials toward rationally designing nanoelectrodes for MSCs with high-energy performance. In comparison with the conventional HMs, the HAN as a kind of microminiaturized HMs has the highest cell density, lower relative density, higher surface area enhancement factor, and excellent compressive performance. With such robust HAN as the keel, the as-prepared nanoelectrodes could possess vertically aligned and robustly stable nanoporous structure. And in contrast to nanoelectrodes design based on nanowires and nanotubes, there is no limit about the aspect ratio to sustain such vertically aligned nanoporous structure. Besides high surface area to provide abundant electroactive sites for surface Faradaic reactions, the vertically aligned nanoporous structure affords the smooth channels to favor the permeability for electrolyte

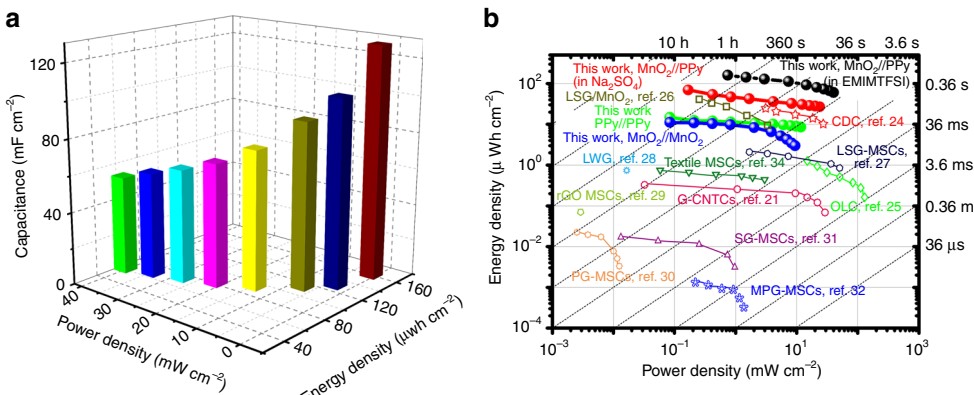

**Fig. 5 Electrochemical performance of HAN@SnO₂@MnO₂//HAN@SnO₂@PPy asymmetric MSCs. a** Typical CV curves of HAN@SnO₂@MnO₂ and HAN@SnO₂@PPy based symmetric MSCs, respectively, with 1.0 M Na₂SO₄ electrolyte. **b** CV curves within different potential ranges. **c** CV curves at different scan rates. **d** GCD profiles at different current densities. **e** Device areal capacitance as a function of current densities. **f** Cycling stability test at a scan rate of 20 mA cm⁻². **g** CV curves at different scan rates, **h** GCD profiles at different current densities, and **i** rate performance of asymmetric MSCs with EMIM-TFSI electrolyte, respectively.

**Fig. 6 Energy performance of HAN@SnO₂@MnO₂//HAN@SnO₂@PPy asymmetric MSCs. a** Areal performance metrics of HAN@SnO₂@MnO₂// HAN@SnO₂@PPy asymmetric MSCs with EMIM-TFSI ionic liquid electrolyte. **b** Ragone plots of MSCs with HAN-based nanoelectrodes compared with some reported MSCs[21,24-34].

infiltration and ion diffusion to access the surface of pseudocapacitive materials, guaranteeing a high utilization efficiency of the pseudocapacitive materials. Taken together, the HAN-based nanoelectrodes design synergizes the effects of both effective ion migration and high electroactive surface area, thus enabling high and reversible capacitive behavior even at high charge–discharge rates. In the current HAN-based nanoelectrodes design, four strategies have been combined together into single MSCs aiming to achieve high-energy storage capability: (i) the utilization of pseudocapacitive materials with high specific capacitance in both positive and negative electrodes; (ii) the adoption of asymmetric device configuration to extend the operating potential window of MSCs; (iii) the application of ionic liquid electrolyte to further widen the operating potential window of MSCs; and more importantly, (iv) the prominent nanoelectrode architecture design based on HAN for both positive and negative electrodes. The combination of pseudocapacitive materials with unique nanoelectrodes architecture results in the increased areal capacitance with reduced footprint, meanwhile the adoption of asymmetric device configuration coupled with ionic liquid electrolyte helps to overcome the narrow operating potential window of pseudocapacitive materials. As a result, the asymmetric MSCs constructed with the HAN-based nanoelectrodes exhibit one of the most remarkable comprehensive areal device performance metrics.

We emphasize that nanoelectrode design with the robust HAN as the keel to assemble electroactive materials inherits the unique structural characteristics of the HMs and can be manipulated to achieve desired application requirements for electrochemical devices beyond MSCs. Our work opens up the ample opportunities for further expanding the application range of the HMs and also provides a paradigm about rationally designing nanostructures for various functional devices with new features and high performance.

## Methods

**Preparation of HAN**. The HAN was fabricated through a nanoindentation-anodization process, followed by a precisely controlled two-step chemical etching process. Firstly, a hexagonal patterns was generated onto the surface of aluminum foil by nanoindentation, then the aluminium foil was anodized in a 0.4 M phosphoric acid ($H_3PO_4$) aqueous solution at 160 V for different time. The temperature of the solution and the aluminium foil was kept at 10 °C during the anodization process. After anodization, the unanodized aluminum was etched by a mixture aqueous solution of $CuCl_2$ (85 wt%) and HCl (15 wt%). Thereafter a two-step chemical etching process was applied to obtain the HAN: (1) etching with a 5 wt% $H_3PO_4$ solution at 60 °C to achieve an open-end membrane; (2) further etching with a 5 wt% $H_3PO_4$ solution at 30 °C to finally obtain HAN. The surface area enhancement factor ($A$) for HAN was calculated by Eq. 1:

$$A = \frac{\frac{6}{\sqrt{3}} \times d \times h}{s^2 \times \sin\theta} \quad (1)$$

where $d$ and $h$ are corresponding to the cell diameter and depth of the cell, respectively, $s$ is the intercell distances, measured between the centers of the cells, and $\theta$ is the angle of the pattern in which the cells are positioned. For the typical HAN structure with cell depth of 25 μm, the specific values are $d = 384$ nm, $h = 25$ μm, $s = 400$ nm and $\theta = 60°$, respectively. Accordingly, the $A$ is calculated to be 240.

**Fabrication of HAN-based nanoelectrodes**. The HAN@SnO$_2$ substrate was produced by conformally depositing SnO$_2$ on the surface of the as-prepared HAN at 250 °C in an ALD system (Picosun, SUNALE R-150). The precursors were tin (IV) chloride (SnCl$_4$) for tin and ultrapure water for oxygen, respectively, and high-purity nitrogen was the carrier and purging gas. Specifically, each ALD cycle consisted of a 0.2 s pulse of SnCl$_4$ and a 4 s purge of N$_2$, followed by a 1 s pulse of H$_2$O and an 8 s purge of N$_2$, and this procedure was repeated 1500 times. The growth rate of SnO$_2$ is estimated to be about 0.16 Å per cycle. With the as-obtained HAN@SnO$_2$ as the working electrodes, the HAN-based nanoelectrodes were fabricated through electrochemical deposition method. The HAN@SnO$_2$@PPy electrode was prepared by electrochemical polymerization of PPy onto HAN@SnO$_2$ substrate. The plating solution consisted of 0.1 M pyrrole monomer (98%) and 0.2 M oxalic acid, and the applied potential was 0.8 V (vs. Ag/AgCl)[37]. Likewise, the HAN@SnO$_2$@MnO$_2$ electrode was fabricated by electrochemical deposition of MnO$_2$ onto the HAN@SnO$_2$ substrate by using an electrolyte consisting of 50 mM

manganese acetate and 100 mM sodium acetate with a constant current density of 1 mA cm$^{-2}$. The active mass of the electrodes were determined according to Faraday's law and 100% charge efficiency was assumed, according to Eq. 2:

$$m = \frac{QM}{zF} \quad (2)$$

where $Q$ is the charge passed during the electrochemical deposition process of active materials (i.e., PPy/MnO$_2$). And it is worth noting that the charge is precisely adjust here in order to achieve the same layer thickness of active materials for HAN@SnO$_2$ with different cell depth. $M$ is the molar mass of the active electrode material ($M_{MnO2} = 86.9$ g mol$^{-1}$; $M_{Py} = 65.09$ g mol$^{-1}$), $F$ the Faraday constant and $z$ the number of transferred electrons per active electrode atom ($z = 2$ for MnO$_2$ and PPy deposition).

**Apparatus for characterizations**. The surface morphologies and microstructures of the electrodes were characterized using scanning electron microscopy (ZEISS AURIGA) equipped with an EDX detector. Cyclic voltammetry (CV), galvanostatic charge–discharge (GCD), electrochemical impedance spectroscopy (EIS) measurements, and cycling performance measurements were conducted with a Potentiostat (BioLogic, VSP). Nanoindentation was conducted with the Triboindenter (Hysitron) equipped with a Berkovich diamond indenter with an approximately 600-nm-radius.

**Electrochemical measurements and calculation of the electrochemical performances**. All the electrochemical measurements were carried out in a two-electrode configuration with 1.0 M Na$_2$SO$_4$ aqueous electrolyte and a glass microfiber filter (Whatman, GF/B) as separator as well as 1-ethyl-3-methylimidazolium bis(trifluoromethylsulfonyl)imide (EMIM-TFSI) ionic liquid electrolyte (1 M in acetonitrile). The CV curves were collected with various scan rates of 5 to 2000 mV s$^{-1}$, and the GCD curves were obtained at current densities of 0.2, 0.5, 1, 2, 5, 10, 15, 20 and 25 mA cm$^{-2}$. The EIS measurement was performed with a frequency range from 100 MHz to 10 mHz and a 5 mV AC amplitude.

The capacitance of the device $C_{GCD}$ (mF cm$^{-2}$) is determined from the GCD profiles based on Eq. 3:

$$C_{GCD} = \frac{I \times \Delta t}{A \times \Delta V} \quad (3)$$

where $\Delta t$ (s) is the discharge time, $I$ (mA) is the discharge current, $A$ (cm$^2$) is total footprint area of device, and $\Delta V$ (V) is the voltage window.

The device capacitance $C_{CV}$ (mF cm$^{-2}$) can also be calculated from CV curves based on Eq. 4:

$$C_{CV} = \frac{\int I dV}{v \times \Delta V \times A} \quad (4)$$

where $I$ (A) is the response current and $v$ (V s$^{-1}$) is the potential scan rate.

With GCD plots, the energy density ($E$, unit: μWh cm$^{-2}$) and power density ($P$, unit: mW cm$^{-2}$) of the device were calculated by Eqs. 5 and 6:

$$E = 0.5 \times C \times \Delta V^2 \quad (5)$$

$$P = \frac{E}{\Delta t} \quad (6)$$

where $C$ (mF cm$^{-2}$) is the areal capacitance of the device, $\Delta t$ (s) is the discharge time.

## Date availability

The data that support the findings of this study are available from the corresponding authors on reasonable request.

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

## Acknowledgements

The authors gratefully acknowledge the support from German Research Foundation (DFG: LE 2249/4-1 and LE 2249/5-1), as well as financial support for Long Liu from the China Scholarship Council (CSC).

## Author contributions

Zh. L and Lo. L contributed equally to this work. Y.L. supervised the project. The concept was conceived by H.Z. and Y.L. Zh. L, Lo. L and H.Z. were involved in the fabrication and related characterizations. Zh. L, Lo. L, S.C. and Le. L performed the electrochemical experiments and data analysis under the guidance of H.Z., F.L., Zh. L, J.K. and Y.L. H.Z. and Y.L. wrote the paper with the assistance of F.L., Y.Z. and J.K.

## Competing interests

The authors declare no competing interests.
