## [Peer Review File · Nature Communications]

Reviewers' comments:

Reviewer #1 (Remarks to the Author):

This manuscript presented an interesting design strategy of micro-supercapacitor nanoelectrodes based on honeycomb alumina nanoscaffolds. Ultrathin alumina honeycomb nanoscaffold has been successfully used as the framework, using which the designed nanoelectrode has stable and vertically-aligned nanopores with large ion-accessible surface area and low ion transport resistance, as well as a large scaffold/electrode proportion due to the ultrathin walls. The device was shown an outstanding electrochemical performance in terms of peak energy and power density. The work should be worthwhile publishing after addressing the main concerns as listed below:

1. Mechanical stability was less discussed, needs more elaboration. For example, more discussions should be provided about the electrochemical performance of HAN@SnO₂ as current collectors under mechanical pressing, say in Figure 2g. What are the residues in Figure 2i? Is it the residual glass fibers?
2. Why were MnO₂ and PPy selected as the electrode materials, any particular justifications or reasons?
3. How about the performance cycling stability of the symmetric micro-supercapacitors? Beside morphological changes of electrode after long-termed cycling (Supplementary Figure 9), anything else?
4. More detailed discussion could be provided about the effect of the charge transport resistance variations in HAN@SnO₂@MnO₂ and HAN@SnO₂@PPy electrodes on the electrochemical energy storage performance.
5. In Figures 5b-c, the CV curves of the asymmetric micro-supercapacitors have retained the rectangular shape, meaning the good electrode capacity matching, which should be pointed out in detail.
6. In Figure 5g, a distorted CV and GCD curves was observed on the asymmetric micro-supercapacitors with ionic liquid electrolyte, as compared to those with Na₂SO₄ aqueous electrolyte, any explanations?

Reviewer #2 (Remarks to the Author):

Comments to Authors

Microminiaturized honeycomb monoliths have been widely exploited as catalyst supporters in different gaseous reactor applications such as chemical and refining processes, catalytic combustion, ozone abatement, and photocatalytic air purification. This article reports the fabrication of microminiaturized honeycomb alumina nano scaffolds as a robust nanostructuring platform to assemble active materials for micro-supercapacitors. It's a smart and interesting strategy to fabricate the microminiaturized HANs. However, the insulating HANs are not preferable for assembling active materials of micro-supercapacitors but could be better for some catalysis. Moreover, the electrochemical performances of those devices in this paper are not impressive. As such, I feel that this work would find a more suitable outlet in another journal.

Some suggestions for authors to improve the quality of this paper:

1. What's the electrolyte in HAN@SnO₂//HAN@SnO₂ cell? SnO₂ is not a good active material for supercapacitor.
2. How to get the discharge density in Fig. 2c? The current density isn't linear at a scan rate of 20 V/s.
3. What's the weight-specific capacitance of HAN@SnO₂@PPy on the HAN? The pore depth of HAN in both two nanoelectrodes was 25 μm, and the mass loading of MnO₂ and PPy would be high.
4. It's double-layer but not pseudocapacitive mechanism in the HAN@SnO₂@PPy according to the CVs in Fig. 3 since the polypyrrole (PPy) electrode could not react with Na₂SO₄ electrolyte.
5. Very confusing, why replace aqueous electrolyte with the ionic liquid electrolyte EMIM-TFSI for

HAN@SnO₂@MnO₂//HAN@SnO₂@PPy cell. Its coulombic efficiency is also low in an enlarged potential range of 0 – 3.0 V (Fig. 5).

Point-by-point reply to the comments of the reviewers

Reviewer #1

This manuscript presented an interesting design strategy of micro-supercapacitor nanoelectrodes based on honeycomb alumina nanoscaffolds. Ultrathin alumina honeycomb nanoscaffold has been successfully used as the framework, using which the designed nanoelectrode has stable and vertically-aligned nanopores with large ion-accessible surface area and low ion transport resistance, as well as a large scaffold/electrode proportion due to the ultrathin walls. The device was shown an outstanding electrochemical performance in terms of peak energy and power density. The work should be worthwhile publishing after addressing the main concerns as listed below.

Reply: We greatly appreciate the reviewer for the positive feedbacks on the contents presented in our manuscript and the support on the publication of this work.

Question 1: *Mechanical stability was less discussed, needs more elaboration. For example, more discussions should be provided about the electrochemical performance of HAN@SnO₂ as current collectors under mechanical pressing, say in Figure 2g. What are the residues in Figure 2i? Is it the residual glass fibers?*

Reply: We thank the reviewer for the useful comment and suggestion. Following your suggestion, we have provided more discussion on the electrochemical characterizations of HAN@SnO₂ as nanostructured current collectors under mechanical pressing in the revised manuscript (please see the revised manuscript, Pages 10-11). The residues in Figure 2i of the manuscript is the residual glass fibers from a glass microfiber filter (Whatman, GF/B) that was used as the separator to assemble the HAN@SnO₂//HAN@SnO₂ device. Partials of the glass microfiber filter were damaged and fallen off during the disassembly of the HAN@SnO₂//HAN@SnO₂ device for SEM characterizations.

Question 2: *Why were MnO₂ and PPy selected as the electrode materials, any particular justifications or reasons?*

Reply: Thank you very much for the thoughtful comment. In the present work, MnO₂ and PPy are only selected as the examples of electrode active materials to verify the feasibility of utilizing HAN@SnO₂ as nanostructured current collectors to produce nanostructured electrodes for supercapacitors, and they can be feasibly replaced with other electrode active materials for supercapacitors, such as RuO₂, Fe₃O₄, V₂O₅, Nb₂O₅, PEDOT, PANI, etc.

***Question 3:** How about the performance cycling stability of the symmetric micro-supercapacitors? Beside morphological changes of electrode after long-termed cycling (Supplementary Figure 9), anything else?*

Reply: We thank the reviewer for the insightful comments. The cycling stability of the symmetric micro-supercapacitors (MSCs) was evaluated by continuously charging and discharging MSCs at a constant current density of 20 mA cm⁻². It is observed from Figure R1 that HAN@SnO₂@MnO₂//HAN@SnO₂@MnO₂ symmetric MSCs retain 92% capacitance retention after 30,000 continued charge-discharge cycles, and likewise, HAN@SnO₂@PPy//HAN@SnO₂@PPy symmetric MSCs preserve 94% capacitance retention under the same conditions. The high capacitance retention at a high current density highlights the significant role of robust HAN@SnO₂-based nanoelectrodes in rendering the fast and continuous ion diffusion toward the electrode active materials, finally leading to the extraordinary long cycle life. Besides, it should be noted that there are slightly morphological changes at the top of HAN@SnO₂-based nanoelectrode after long-termed cycling (Supplementary Figure 9), which is believed to be the reason for the slight performance degradation. Apart from that, there were almost no significant structural or morphological changes, indicating the remarkable structural stability of HAN@SnO₂-based nanoelectrodes. Figure R1 has been included in Supplementary Figure 9 of the revised supplementary information (please see the revised supplementary information, Page 10).

Figure R1. The cyclic performance of HAN@SnO₂@MnO₂//HAN@SnO₂@MnO₂ symmetric MSCs and HAN@SnO₂@PPy//HAN@SnO₂@PPy symmetric MSCs, respectively, measured at the current density of 20 mA cm⁻² for 30,000 continued charge-discharge cycles.

Question 4: More detailed discussion could be provided about the effect of the charge transport resistance variations in HAN@SnO₂@MnO₂ and HAN@SnO₂@PPy electrodes on the electrochemical energy storage performance.

Reply: We appreciate the reviewer for giving this useful suggestion. Following your suggestion, we have provided some more discussions about the electrochemical impedance spectroscopy (EIS) properties of MSCs to identify the kinetics of electron and ion transport within both HAN@SnO₂@MnO₂//HAN@SnO₂@MnO₂ and HAN@SnO₂@PPy//HAN@SnO₂@PPy MSCs as well as the effect on the electrochemical energy storage performance (please see the revised manuscript, Pages 14-16).

Question 5: In Figures 5b-c, the CV curves of the asymmetric micro-supercapacitors have retained the rectangular shape, meaning the good electrode capacity matching, which should be pointed out in detail.

Reply: We highly appreciate the suggestion from the reviewer. The balance of the electrode masses or charges is crucial for constructing asymmetric supercapacitors. The charges passed through the positive and negative electrodes in an asymmetric supercapacitor must be the same. In this case, the masses of the positive and negative electrodes are fixed according to the charges passed through as demonstrated in equation

$$Q^- = m^- Q_{sp}^- = m^- C_{sp}^- \Delta E^- = m^+ C_{sp}^+ \Delta E^+ = m^+ Q_{sp}^+ = Q^+$$

where Q is the charge of the electrode, m is the mass of the electrode, C_{sp} is the specific capacitance of the electrode, ΔE is the potential range, and the superscript + and – represent the positive and negative electrodes, respectively. In our work, it is worth noting that the C_{sp} of both the positive and negative electrodes were calculated based on a two-electrode symmetric MSCs rather than a conventional three-electrode configuration cell, which helps to reach the good electrode capacity matching in an asymmetric MSCs.

Question 6: *In Figure 5g, a distorted CV and GCD curves was observed on the asymmetric micro-supercapacitors with ionic liquid electrolyte, as compared to those with Na₂SO₄ aqueous electrolyte, any explanations?*

Reply: Thank you very much for the insightful comment. The large ion size of the ionic liquids causes the high viscosity and large charge transfer resistance of the electrolytes, thus leading to the distorted CV and GCD curves (*ACS Nano* **2013**, 7, 6899; *Energy Environ. Sci.* **2013**, 6, 1623.). Accordingly, we have modified the related discussions in the revised manuscript (please see the revised manuscript, Page 19).

Reviewer #2:

Microminiaturized honeycomb monoliths have been widely exploited as catalyst supporters in different gaseous reactor applications such as chemical and refining processes, catalytic combustion, ozone abatement, and photocatalytic air purification. This article reports the fabrication of microminiaturized honeycomb alumina nano scaffolds as a robust nanostructuring platform to assemble active materials for micro-supercapacitors. It's a smart and interesting strategy to fabricate the microminiaturized HANs. However, the insulating HANs are not preferable for assembling active materials of micro-supercapacitors but could be better for some catalysis. Moreover, the electrochemical performances of those devices in this paper are not impressive. As such, I feel that this work would find a more suitable outlet in another journal.

Reply: We thank the reviewer for the time to review our manuscript.

Firstly, we appreciate the reviewer for the positive evaluation of our work regarding “..... microminiaturized honeycomb alumina nanoscaffolds as a robust nanostructuring platform to assemble active materials for micro-supercapacitors. It's a smart and interesting strategy.....”.

For the comment “*the insulating HANs are not preferable for assembling active materials of micro-supercapacitors*”, we agree that the as-prepared HANs are insulating (we also mentioned it in our manuscript). However, in order to enable our HANs to be preferable for supercapacitor applications, we coated the HANs with a thin layer of SnO₂ and hence converting the insulating HANs to nanostructured current collectors (denoted as HAN@SnO₂ in our manuscript) for assembling electrode active materials of micro-supercapacitors. SnO₂ was coated by atomic layer deposition (ALD), by which a high quality (conformally and uniformly) coated layer is ensured. This ALD-coated SnO₂ has a much higher electrical conductivity than those of many other metallic oxides. Thereafter, the HAN@SnO₂ hybrid structure has been further verified to be feasible as a nanostructured current collector to assemble MnO₂ and PPy (active supercapacitor materials) as electrodes. In another word, the robust nanostructuring platform in our work to assemble active materials for micro-supercapacitors is HAN@SnO₂ rather than HAN, which is an important aspect of our work.

Regarding the comment on the performance of micro-supercapacitors “*the electrochemical performances of those devices in this paper are not impressive*”, we are sorry that we could not agree with this comment based on the following:

As known, the development of miniaturized energy storage devices is one of the main technological challenges for the implementation of Internet of Things (IOTs). Micro-supercapacitor (MSC) is an attractive solution to fulfil the energy requirements of autonomous, smart, maintenance-free and miniaturized IOT devices, however, restrict from their insufficient energy. Currently, there have been intensive efforts in tackling the deficient energy issue of MSCs. Given that many IOTs applications are constrained by area, the areal performance metrics in terms of the device capacitance, energy and power densities normalized to the footprint of MSCs are now recognized as one of the most reasonable evaluation criteria to MSCs as miniaturized energy sources for micro-devices (*Nat. Nanotechnol.* **2017**, *12*, 7; *Energy Environ. Sci.* **2014**, *7*, 867; *J. Electrochem. Soc.* **2017**, *164*, A1487; *Adv. Mater.* **2019**, 1805864; *Adv. Mater.* **2019**, 1900583.). Therefore, nanoelectrode design represents an important direction in the field of MSCs, aiming to improving energetic performance but still keeping a small footprint size. Even though the nanoelectrode design strategies for MSCs have progressed in recent years (*Nat. Nanotechnol.* **2017**, *12*, 7; *Small Methods* **2019**, *3*, 1800367; *Energy Environ. Sci.* **2019**, *12*, 96.), the areal energy density of MSCs still needs to be further improved. In the present work, we demonstrate the feasibility of the HAN-based nanoelectrode design strategy for achieving MSCs with high areal energy and power performance. To the best of our knowledge, the MSCs in this work is among the best

comprehensive energetic performance of the reported MSCs (please see Figure 6b in the revised manuscript and Table S1 in the revised supplementary information). Particularly, the peak energy density of our MSCs reaches $160 \mu\text{Wh cm}^{-2}$, which is about fourfold that of the representative MSCs based on carbide-derived-carbons (of about $40 \mu\text{Wh cm}^{-2}$) but with a similar peak power density (*Science* **2016**, 351, 691.). Moreover, the areal energy density of our MSCs is even comparable with those of the state-of-the-art three-dimensional micro-batteries (*Nat. Commun.* **2013**, 4, 1732; *Proc. Natl. Acad. Sci. U.S.A.* **2015**, 112, 6573) but with much higher areal power density. Up to now (about six months after we submitted this manuscript on April 23rd), the comprehensive energetic performance of our MSCs is still the best one (as shown in Figures R2 and R3, and all the corresponding data are included in the revised Supplementary Table 1). Not limited to MSCs, the HAN-based nanoelectrode concept shown in this work shall be applicable to assemble different catalysts for various catalysis applications, as pointed out by the reviewer. Therefore, we believe that the very good performance of our MSCs will be of high interest to the scientists working in the field of electrochemical energy storage and to the broad readership of *Nature Communications*.

Figure R2. Radar plot of the comprehensive performance of our MSC in comparison with those of MSCs reported in 2019.

Figure R3. The areal energy density of our MSC in comparison with those of the MSCs reported in 2019.

Question 1: What's the electrolyte in HAN@SnO₂//HAN@SnO₂ cell? SnO₂ is not a good active material for supercapacitor.

Reply: We appreciate the reviewer for this comment. The electrolyte in the HAN@SnO₂//HAN@SnO₂ cell is 1.0 M Na₂SO₄ aqueous solution.

We totally agree with the reviewer that SnO₂ is not a good active material for supercapacitor. Actually in this work, we used SnO₂ only as the conductive layer to convert the insulating HAN to current collector (as mentioned above), rather than as an electrode active material for supercapacitors. The obtained HAN@SnO₂ was served as a nanostructured current collector for assembling electrode active materials (*i.e.*, MnO₂ and PPy in this work) of supercapacitors.

Question 2: How to get the discharge density in Fig. 2c? The current density isn't linear at a scan rate of 20 V/s.

Reply: Thanks for the comment. The discharge current densities in Fig. 2c of the manuscript were calculated by normalizing the discharge currents (taken from the cyclic voltammetry

profiles at 0.5 V for discharge segments) to the footprint area of HAN@SnO₂//HAN@SnO₂ cell (*Adv. Energy Mater.* **2015**, 5, 1500003.). The nonlinear current density at a scan rate of 20 V s⁻¹ for the HAN@SnO₂//HAN@SnO₂ cell should be due to the limited ion diffusion at higher scan rates (*Adv. Energy Mater.* **2015**, 5, 1500003) and the relatively low electrical conductivity of SnO₂ compared to metals (*Adv. Sci.* **2015**, 3, 1500299; *Adv. Energy Mater.* **2019**, 9, 1901061).

Question 3: *What's the weight-specific capacitance of HAN@SnO₂@PPy on the HAN? The pore depth of HAN in both two nanoelectrodes was 25 μm, and the mass loading of MnO₂ and PPy would be high.*

Reply: Many thanks for the comment. We would like to point out that all the reported capacitances in our manuscript are the areal device capacitances rather than the areal electrode capacitances because the areal performance metrics in terms of the device capacitance, energy and power densities (normalized to the footprint area of micro-supercapacitors) are now recognized as one of the most reasonable evaluation criteria to micro-supercapacitors as miniaturized energy sources for micro-devices (*Nat. Nanotechnol.* **2017**, 12, 7; *Energy Environ. Sci.* **2014**, 7, 867; *J. Electrochem. Soc.* **2017**, 164, A1487; *Adv. Mater.* **2019**, 1900583.). The MnO₂ mass loading was 0.65 mg cm⁻² for HAN@SnO₂@MnO₂ electrodes consisting of 25-μm-pore-deep HAN, while the PPy mass loading was 1.32 mg cm⁻² for HAN@SnO₂@PPy electrodes consisting of 25-μm-pore-deep HAN. According to Supplementary Figures 6c and 6f, the calculated weight specific capacitances are 372.3 F g⁻¹ for MnO₂ at a scan rate of 0.2 mA cm⁻² (or 0.31 A g⁻¹) and 239.4 F g⁻¹ for PPy at a scan rate of 0.2 mA cm⁻² (or 0.15 A g⁻¹), respectively.

Question 4: *It's double-layer but not pseudocapacitive mechanism in the HAN@SnO₂@PPy according to the CVs in Fig. 3 since the polypyrrole (PPy) electrode could not react with Na₂SO₄ electrolyte.*

Reply: We thank the reviewer for the comment, but feel sorry again that we could not fully agree with this. Conductive polymer polypyrrole (PPy) is actually a kind of pseudocapacitor electrode materials (*J. Electrochem. Soc.* **2002**, 149, A1058; *Adv. Mater.* **2013**, 25, 591; *Nano Lett.* **2014**, 14, 2522; *Adv. Funct. Mater.* **2015**, 25, 4626; *Nano Energy* **2016**, 22, 422; *Natl. Sci. Rev.* **2017**, 4, 71; *Chem. Soc. Rev.* **2017**, 46, 6816; etc.), because the electron delocalization in π-orbital conjugation along the conductive polymer backbone gives the compounds the ability to be oxidized or reduced (*Energy Environ. Sci.* **2010**, 3, 1238). The

pseudocapacitive properties of PPy have been well studied (*Synth. Met.* **1993**, *55*, 1329; *Polymer* **1991**, *32*, 1354; *Synth. Met.* **1999**, *101*, 335; *Phys. Chem. Chem. Phys.* **2014**, *16*, 3523; etc.). Different from pseudocapacitive metal oxides and electric double layer capacitive materials, the conducting polymer has a unique energy storage process. Rudge *et al.* first reported that the charge storage and release of PPy occurs through p-doping and p-dedoping by electrolytes (*J. Power Sources*, **1994**, *47*, 89.). When an external potential is applied to the PPy electrodes, electrons are abstracted from the PPy backbone while anions (which are SO_4^{2-} ions in our case) are incorporated from the electrolyte (here is 1.0 M Na_2SO_4 aqueous solution in present work) into the surface of PPy to keep the charge balance. Moreover, it is worth noting that all the electrochemical measurements (*i.e.*, cyclic voltammetry, galvanostatic charge-discharge, and electrochemical impedance spectroscopy) were carried out in a two-electrode configuration rather than a three-electrode configuration. The shape of cyclic voltammetry profiles measured from a two-electrode configuration with well-matched capacity of positive and negative electrodes would be more symmetric.

Question 5: *Very confusing, why replace aqueous electrolyte with the ionic liquid electrolyte EMIM-TFSI for HAN@SnO₂@MnO₂//HAN@SnO₂@PPy cell. Its coulombic efficiency is also low in an enlarged potential range of 0 – 3.0 V (Fig. 5).*

Reply: Thank for this comment. Most of the recent research efforts are devoted to improving the energy density of supercapacitors, and the replacement of aqueous electrolytes with ionic liquid electrolytes is regarded as an effective strategy to extend the working potential window of supercapacitors and thus improve the energy density of supercapacitors. The main purpose to replace aqueous electrolyte with the ionic liquid electrolyte is to further extend the working potential window of HAN@SnO₂@MnO₂//HAN@SnO₂@PPy MSCs, and subsequently to improve the energy density of devices. With 1.0 M Na_2SO_4 aqueous electrolyte, the maximum potential window of HAN@SnO₂@MnO₂//HAN@SnO₂@PPy MSCs is 1.6 V, and it can be further extended to 3.0 V when using the EMIM-TFSI ionic liquid electrolyte. Although the device capacity was slightly reduced from 144 mF cm^{-2} to 128 mF cm^{-2} at a same current density of 0.5 mA cm^{-2} , the overall device energy density of HAN@SnO₂@MnO₂//HAN@SnO₂@PPy MSCs with the EMIM-TFSI ionic liquid electrolyte is significantly enhanced attributing to the extended working potential window (*i.e.*, from 1.6 V to 3.0 V) according to the following equation:

$$E = \frac{1}{2} CV^2$$

where E ($\mu\text{Wh cm}^{-2}$) is the energy density, C (mF cm^{-2}) is the device capacity, and V (V) is the working potential window of the device, respectively.

Moreover, the distorted galvanostatic charge-discharge profiles of MSCs with ionic liquid electrolytes compared to those of MSCs with aqueous electrolytes don't suggest the low Coulombic efficiency. For example, symmetric MSCs assembled with two same $\text{Cu(OH)}_2@FeOOH/\text{Cu}$ electrodes also exhibited distorted galvanostatic charge-discharge profiles in an extended potential window of 0 – 1.5 V when utilizing EMIMBF_4 ionic liquid electrolytes, but the Coulombic efficiency of MSCs still kept nearly 100% after being charged and discharged 10,000 cycles (*Energy Environ. Sci.* **2019**, *12*, 194.). Similarly in the present work, the $\text{HAN@SnO}_2@MnO_2//\text{HAN@SnO}_2@PPy$ MSCs with the EMIM-TFSI ionic liquid electrolytes still exhibit remarkable cycling stability with 82.5% of initial capacitance at 20 mA cm^{-2} over a potential window of 0 – 3.0 V withstanding continued 10,000 charge-discharge cycles and also keeping a high Coulombic efficiency of nearly 100% (as shown in Supplementary Figure 10b of the revised supplementary information).

REVIEWERS' COMMENTS:

Reviewer #1 (Remarks to the Author):

The authors have fully addressed the concerns from reviewers by providing careful and considerate responses, and provided additional/necessary experimental and reference comparison data as well as the necessary revision of the manuscript and supplementary materials. Therefore the manuscript can be accepted as it is.

Reviewer #2 (Remarks to the Author):

I have reviewed the authors' responses to my review comments as well as those of the other reviewers, and believe they have satisfactorily addressed the questions raised.

Reviewer #1

Comments: *The authors have fully addressed the concerns from reviewers by providing careful and considerate responses, and provided additional/necessary experimental and reference comparison data as well as the necessary revision of the manuscript and supplementary materials. Therefore the manuscript can be accepted as it is.*

Reply: We highly appreciate the reviewer's constructive comments and suggestions to help us improve the quality of our manuscript.

Reviewer #2

Comments: *I have reviewed the authors' responses to my review comments as well as those of the other reviewers, and believe they have satisfactorily addressed the questions raised.*

Reply: We really appreciate the reviewer for reviewing our manuscript, and providing constructive comments and suggestions.